# Robust Pre-Training by Adversarial Contrastive Learning

**Ziyu Jiang[1], Tianlong Chen[2], Ting Chen[3], Zhangyang Wang[2]**

[1]Texas A&M University, [2]University of Texas at Austin, [3]Google Research, Brain Team
jiangziyu@tamu.edu, {tianlong.chen,atlaswang}@utexas.edu, iamtingchen@google.com

## Abstract

Recent work has shown that, when integrated with adversarial training, self-supervised pre-training can lead to state-of-the-art robustness [1]. In this work, we improve robustness-aware self-supervised pre-training by learning representations that are consistent under both data augmentations and adversarial perturbations. Our approach leverages a recent contrastive learning framework [2], which learns representations by maximizing feature consistency under differently augmented views. This fits particularly well with the goal of adversarial robustness, as one cause of adversarial fragility is the lack of feature invariance, i.e., small input perturbations can result in undesirable large changes in features or even predicted labels. We explore various options to formulate the contrastive task, and demonstrate that by injecting adversarial perturbations, contrastive pre-training can lead to models that are both label-efficient and robust. We empirically evaluate the proposed *Adversarial Contrastive Learning* (ACL) and show it can consistently outperform existing methods. For example on the CIFAR-10 dataset, ACL outperforms the previous state-of-the-art unsupervised robust pre-training approach [1] by 2.99% on robust accuracy and 2.14% on standard accuracy. We further demonstrate that ACL pre-training can improve semi-supervised adversarial training, even when only a few labeled examples are available. Our codes and pre-trained models have been released at: https://github.com/VITA-Group/Adversarial-Contrastive-Learning.

## 1 Introduction

Label efficiency and model robustness are two desirable characteristics when it comes to building machine learning models, including the training of deep neural networks. Traditional supervised learning of deep networks requires a lot of labeled data, whose annotation costs can be prohibitively high. Utilizing unlabeled data, e.g., by various unsupervised or semi-supervised learning techniques, is thus of booming interests. One particular branch of unsupervised learning based on contrastive learning has shown great promise recently [3, 4, 5, 6, 7, 2]. The latest contrastive learning framework [2] can largely improves label efficiency of the deep networks, e.g., surpassing the fully-supervised AlexNet [8] when fine-tuned on only 1% of the labels.

The labeling scarcity is even amplified when we come to adversarially robust deep learning [9], i.e., to training deep models that are not fooled by maliciously crafted, although imperceivable perturbations. As suggested in [10], the sample complexity of adversarially robust generalization is significantly higher than that of standard learning. Recent results [11, 12, 13] advocated unlabeled data to be a powerful backup for training adversarially robust models as well, by using unlabeled data to form an auxiliary loss (e.g., a label-independent robust regularizer or a pseudo-label loss).

Only lately, researchers start linking self-supervised learning to adversarial robustness. [14] proposed a multi-task learning framework that incorporates a self-supervised objective to be co-optimized with the conventional classification loss. The latest work [1] introduced adversarial training [15] into self-supervision, to provide general-purpose robust pre-trained models for the first time. However, existing work is based on ad-hoc "pretext" self-supervised tasks, and as shown in [6, 2], the learned

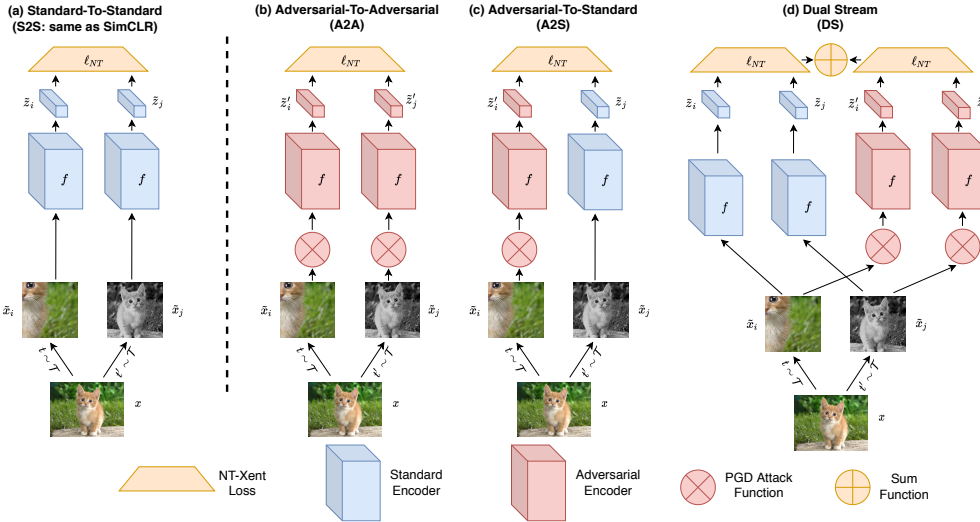

Figure 1: Illustration of workflow comparison: (a) The original SimCLR framework [2], a.k.a., standard to standard (no adversarial attack involved); (b) - (d) three proposed variants of our adversarial contrastive learning framework: A2A, A2S, and DS (our best solution). Note that, whenever more than one encoder branches co-exist in one framework, they by default share all weights, except that adversarial and standard encoders will use independent BN parameters.

unsupervised representations can be largely improved with contrastive learning, a new family of approaches for self-supervised learning.

In order to learn data-efficient robust models, we propose to integrate contrastive learning with adversarially robust deep learning. Our work is inspired by recent success of contrastive learning, where the model is learned by maximizing consistency of data representations under differently augmented views [2]. This fits particularly well with adversarial training, as one cause of adversarial fragility could be attributed to the non-smooth feature space near samples, i.e., small input perturbations can result in large feature variations and even label change.

By properly combining adversarial learning and contrastive pre-training (i.e. SimCLR [2]), we could achieve the desirable feature consistency. The resultant unsupervised pre-training framework, called *Adversarial Contrastive Learning* (**ACL**), is thoroughly discussed in Section 2. As the pre-training plays an increasingly important role for adversarial training [1], by leveraging more powerful contrastive pre-training of unsupervised representations, we further contribute to pushing forward the state-of-the-art adversarial robustness and accuracy.

On CIFAR-10/CIFAR-100, we extensively benchmark our pre-trained models on two adversarial fine-tuning settings: *fully-supervised* (using all training data and their labels) and *semi-supervised* (using all training data, but only partial/few shot labels). ACL pre-training leads to new state-of-the-art in both robust and standard accuracies. For example, in the fully-supervised setting, we obtain **2.99%** robust and **2.14%** standard accuracy improvements, over the previous best adversarial pre-training method [1] on CIFAR-10. For the semi-supervised setting, our approach surpasses the state-of-the-art method [11], by **0.66%** robust and **5.87%** standard accuracy, when 10% labels are available; that gains can become even more remarkable when we go further to extremely low label rates, e.g., 1%.

## 2 Our approach

### 2.1 Preliminaries

**Adversarial Training** Deep networks are notorious for the vulnerability to adversarial attacks [16]. Numerous approaches have been proposed to enhance model adversarial robustness [17, 18, 19, 20, 21, 15, 22, 23, 24, 25]. Among them, Adversarial Training (AT) based methods [15] remain to be the strongest defense option. Define the perturbation $\boldsymbol{\delta}$, using $\ell_\infty$ attack for example:

$$\boldsymbol{\delta} = \arg\max_{\|\boldsymbol{\delta}'\|_\infty \leq \epsilon} \ell(\boldsymbol{\theta}, \boldsymbol{x} + \boldsymbol{\delta}') \tag{1}$$

where $\theta$ represents the parameters of a model. $x$ denotes a given sample. $\ell$ denotes the loss with parameter $\theta$, and the adversarial perturbation $\delta$. AT solves the following (minimax) optimization:

$$\min_{\theta} \mathbb{E}_{x \in \mathcal{D}}(\ell(\theta, x + \delta)) \tag{2}$$

where $\mathcal{D}$ denotes the training set. The standard training (ST) could be viewed as a special case $\epsilon = 0$.

**Unsupervised Adversarial Training**  Among the few existing works [11, 12] utilizing unlabeled data, we introduce the Unsupervised Adversarial Training (UAT) algorithm [11] which yields the state-of-the-art performance so far. The authors introduce three variants: (i) UAT with Online Targets, which exploits a smoothness regularizer on the unlabeled data to minimize the divergence between a normal sample with its adversarial counter-part; (ii) UAT with Fixed Targets, which first trains a CNN using standard training (no AT applied) and then uses the trained model to supply pseudo labels for unlabeled data, finally combining them with labeled data towards AT altogether; and (iii) UAT++, which combines both types of unsupervised losses and was shown to outperform either individually.

**Contrastive Pretraining**  We build this work upon SimCLR [2], a simple yet powerful contrastive pretraining framework for unsupervised representation learning. The main idea of SimCLR is to learn representations by maximizing agreements of differently augmented views of the same image. More specifically, its workflow is illustrated in Fig. 1 (a). Considering an unlabeled sample $x$, SimCLR first augments the sample to be $\tilde{x}_i$ and $\tilde{x}_j$, with two separate data augmentation operators sampled from the same family of augmentations $\mathcal{T}$ (practically, the composition of random cropping and random color distortion). The pair $(\tilde{x}_i, \tilde{x}_j)$ are then processed by the network backbone (denoted by the two weight-sharing standard encoders in Fig. 1 (a)), outputting features $(\tilde{z}_i, \tilde{z}_j)$. After passing through a nonlinear projection head (omitted in the plot), the transformed feature pairs are optimized under a contrastive loss $\ell_{NT}$ (NT-Xent) to maximize their agreement. Upon the completion of training, we only keep the encoder part as the pre-trained model for downstream tasks.

## 2.2 Adversarial Contrastive Learning as pre-training: three options

SimCLR can be considered as **Standard-to-Standard (S2S)** contrasting, as both branches are natural images, and adversarial training nor attack is involved. In our proposed method, *Adversarial Contrast Learning* (**ACL**), we inject robustness components into the SimCLR framework, and discuss three candidate algorithms (Fig. 1 (b) - (d)).

**Adversarial-to-Adversarial (A2A)**  The first option is to directly inject two adversarial attacks after random data augmentations. We first generate $(\delta_i, \delta_j)$ as adversarial perturbations corresponding to the augmented samples $(\tilde{x}_i, \tilde{x}_j)$, using the PGD attack algorithm [15], respectively. Here the loss $\ell$ for generating perturbations is the contrastive loss (NT-Xent). In other words, we now generate two views that are adversarially enforced to be dissimilar, on which we learn to enforce consistency. In this way, the contrastive learning is now performed on $(\tilde{x}_i + \delta_i, \tilde{x}_j + \delta_j)$ instead of $(\tilde{x}_i, \tilde{x}_j)$, enforcing the representations to be invariant to cascaded standard-and-adversarial input perturbations. Note that this could also be viewed as performing attacks with random initialization [26].

**Adversarial-to-Standard (A2S)**  The second option we explore is to replace $(\tilde{x}_i + \delta_i, \tilde{x}_j + \delta_j)$ with $(\tilde{x}_i + \delta_i, \tilde{x}_j)$, i.e., alleviating the encoder's workload of standing against two simultaneous attacks.

Intuitive as it might look, there is a pitfall for A2S implementation. In both SimCLR (S2S) and A2A, their two backbones completely share all parameters. In A2S, while we still hope the adversarial and standard encoder branches to share all convolutional weights, it is important to allow both encoders to maintain their own independent batch normalization (BN) parameters.

Why this matters? As observed in [27, 28], the deep network feature statistics of clean samples $\tilde{x}$ and adversarial samples $\tilde{x} + \delta$ can be very distinct. Mixing the two sets and normalizing using one set of parameters will compromise both robust and standard accuracy in image classification. We also verified this in our setting, that using one BN for standard and adversarial features will be detrimental to unsupervised learning too. We thus turn to the dual Batch Normalization (BN) suggested in [27], i.e., one BN for standard encoder branch and the other for adversarial branch, respectively.

During fine-tuning and testing, we by default employ the BN for adversarial branch, since it leads to higher robustness which is our main goal. We also briefly study the effect of using either BN on the fine-tuning performance as well as the pre-trained representations in Sec. 3.2.

---

**Algorithm 1:** Algorithm of Dual Stream (DS) Pretraining

---

**Input :** A set of clean images $\boldsymbol{x}$; Augmentation family $\mathcal{T}$; Network backbone and projection head $f$, $g$;

**Result :** Standard BN parameters $\boldsymbol{\theta}_{bn}$; Adversarial branch BN parameters $\boldsymbol{\theta}_{bn^{adv}}$ ; The rest parameters $\boldsymbol{\theta}$ in $f$ and $g$;

**for** sampled mini-batch $\boldsymbol{x}$ **do**

> Augment $\boldsymbol{x}$ to be $(\tilde{\boldsymbol{x}}_i, \tilde{\boldsymbol{x}}_j)$ with two augmentations sampled from $\mathcal{T}$.
> Generate the corresponding adversarial mini-batch $(\tilde{\boldsymbol{x}}_i + \boldsymbol{\delta}_i, \tilde{\boldsymbol{x}}_j + \boldsymbol{\delta}_j)$ with
>
> $$\boldsymbol{\delta}_i, \boldsymbol{\delta}_j = \underset{\|\delta_i\|_\infty \leq \epsilon, \|\delta_j\|_\infty \leq \epsilon}{\arg\max} \ell_{NT}(f \circ g(\tilde{\boldsymbol{x}}_i + \boldsymbol{\delta}_i, \tilde{\boldsymbol{x}}_j + \boldsymbol{\delta}_j; \boldsymbol{\theta}, \boldsymbol{\theta}_{bn^{adv}}))$$
>
> $\ell = \ell_{NT}(f \circ g(\tilde{\boldsymbol{x}}_i, \tilde{\boldsymbol{x}}_j; \boldsymbol{\theta}, \boldsymbol{\theta}_{bn})) + \alpha \ell_{NT}(f \circ g(\tilde{\boldsymbol{x}}_i + \boldsymbol{\delta}_i, \tilde{\boldsymbol{x}}_j + \boldsymbol{\delta}_j; \boldsymbol{\theta}, \boldsymbol{\theta}_{bn^{adv}}))$
> Update parameters $(\boldsymbol{\theta}_{bn}, \boldsymbol{\theta}_{bn^{adv}}, \boldsymbol{\theta})$ to minimize $\ell$.

**end**

---

**Dual Stream (DS)**   The third option we explore is to combine S2S and A2A in one. Specifically, for each input $\boldsymbol{x}$, we augment it into twice (creating four augmented views): $(\tilde{\boldsymbol{x}}_i, \tilde{\boldsymbol{x}}_j)$ by standard augmentations (as in S2S), and $(\tilde{\boldsymbol{x}}_i + \boldsymbol{\delta}_i, \tilde{\boldsymbol{x}}_j + \boldsymbol{\delta}_j)$ further with adversarial perturbation (as in A2A).

Our final unsupervised loss consists of a contrastive loss term on the former pair (through two standard branches) and another contrastive loss term on the latter pair (through two adversarial branches). The two terms are by default equally weighted ($\alpha = 1$ in Algorithm. 1). The four branches share all convolutional weights except BN: the two standard branches use the same set of BN parameters, while the two adversarial branches have their one set of BN parameters too.

Compared to A2A which aggressively demands minimizing the "worst-case" consistencies, the final dual-stream loss is aligned with the classical AT's objective [15], i.e., balancing between standard and worst-case (robust) loss terms. The algorithm of DS can be found in Algorithm. 1. We experimentally verify that Dual-Stream is the best among the three ACL variants. A comprehensive comparison of the three variants is presented in Section 3.2.

### 2.2.1   Towards supervised fine-tuning and semi-supervised training

For each of the three variants, we will use their pre-trained network weights as the initialization for subsequent fine-tuning. For the fully supervised setting, we identically follow the adversarial fine-tuning scheme described in [29].

For semi-supervised training, we employ a three-step routine: *i)* we first perform ACL to obtain a pre-trained model (denoted as *PM*) on all unlabeled data ; *ii)* we next adopt *PM* as the initialization to perform standard training using only labeled data, and use the trained model to generate pseudo-labels for all unlabeled data; *iii)* lastly, we leverage *PM* as initialization again, and perform AT over all data (including unlabeled data and their pseudo labels). The routine in principle follows [11, 30, 12] except the new pre-training step (i). The specific loss that we minimize in step *iii* is:

$$\ell_s = \frac{1}{|N_l| + |N_u|}(\alpha \ell_{CE}(\hat{\boldsymbol{x}}_l, y_l; \boldsymbol{\theta}) + (1 - \alpha)T^2 \ell_{distill}(\hat{\boldsymbol{x}}_l, p_l, T; \boldsymbol{\theta}) +$$
$$T^2 \ell_{\text{distill}}(\hat{\boldsymbol{x}}_u, p_u, T; \boldsymbol{\theta}) + \frac{1}{\lambda}KL(\hat{p}(\boldsymbol{x}; \boldsymbol{\theta}), \hat{p}(\hat{\boldsymbol{x}}; \boldsymbol{\theta}))). \tag{3}$$

Here $y_l$ denotes the ground truth of labeled data $\boldsymbol{x}_l$. $p_l$ and $p_u$ represent the soft logits predicted by the step *ii* trained model, for labeled data $\boldsymbol{x}_l$ and unlabeled data $\boldsymbol{x}_u$ respectively. $\boldsymbol{x}$ denotes all data (both $\boldsymbol{x}_u$ and $\boldsymbol{x}_l$). $[\hat{\boldsymbol{x}}_l, \hat{\boldsymbol{x}}_u, \hat{\boldsymbol{x}}]$ are the adversarial samples of $[\boldsymbol{x}_l, \boldsymbol{x}_u, \boldsymbol{x}]$ generated following [11], respectively. $\ell_{CE}$ and $\ell_{distill}$ are Cross-Entropy (CE) loss and Knowledge Distillation loss (with temperature $T$), respectively.

The first two terms in Eqn. (3) represent a convex combination of CE and distillation losses for labeled data ($\alpha$ is a coefficient $\in (0,1)$). The third term is the distillation loss for unlabeled data. The fourth term is feature-consistency robustness loss for all data (with weight $\frac{1}{\lambda}$) following [11], respectively. $|N_l| + |N_u|$ represent the batch size (including both labeled and unlabeled data).

Table 1: Comparison between Random Initialization (RI), *Selfie* [1] and the proposed ACL (DS) on CIFAR-10. ACL (DS) consistently yields the highest *Standard Testing Accuracy* (TA) and *Robust Testing Accuracy* (RA).

| Method | Random Initialization (RI) | | *Selfie* | | ACL (DS) | |
|---|---|---|---|---|---|---|
| Metric | TA(%) | RA(%) | TA(%) | RA(%) | TA(%) | RA(%) |
| CIFAR-10 | 79.90 | 49.36 | 80.05 | 49.83 | 82.19 | 52.82 |
| CIFAR-100 | 49.12 | 23.07 | 54.63 | 24.75 | 56.77 | 28.33 |

## 3 Experiments and analysis

**Experimental settings:** We evaluate three datasets: CIFAR-10, CIFAR-10-C [31], CIFAR-100. We follow [1] to employ two metrics: 1) *Standard Testing Accuracy (TA)*: The classification accuracy on clean testing images; and 2) *Robust Testing Accuracy (RA)*: the classification accuracy on adversarial perturbed testing images. Evaluation on unforeseen attacks is also considered [32].

For contrastive pre-training, we identically follow SimCLR [2] for all the optimizer settings, augmentation and projection head structure. We choose 512 for batch size and train for 1000 epochs. To generate adversarial perturbations, we use the $\ell_\infty$ PGD attack [15], following all hyperparameters used by [1], except that we only run the PGD for 5 steps in the pre-training stage for faster convergence.

Experiments are organized as following: Section 3.1 first compares our best proposed option: ACL with Dual-Stream (DS), with existing methods, first as pre-training for supervised adversarial fine-tuning (Section 3.1.1); then as part of semi-supervised adversarial training (Section 3.1.2). Section 3.2 then delves into an ablation study comparing the three ACL options: A2A, A2S and DS.

### 3.1 Comparing ACL with state-of-the-arts

#### 3.1.1 ACL pre-training for supervised adversarial fine-tuning

We compare our ACL Dual Stream (DS), with the state-of-the-art robust pre-training [1], on CIFAR-10 and CIFAR-100. For the latter, we choose its single best pretext task, *Selfie*, as the comparison subject (and call it so hereinafter). By default, we use ResNet-18 [33] as the backbone all experiments hereinafter, as was also adopted by [29], unless otherwise specified. ACL (DS) is employed for training the encoder of ResNet-18 (excluding the fully connected (FC) Layer), and then the pre-trained encoder is added with the zero-initialized FC layer to start fine-tuning.

We compare ACL (DS) and *Selfie* by conducting adversarial fine-tuning over their pre-trained models. For the fully-supervised tuning, we employ the loss in TRADE [29] for adversarial fine-tuning, with the regularization weight $1/\lambda$ as 6. The fine-tuned models are selected based on the held-out validation RA. We use SGD with 0.9 momentum and batch size 128. By default, we fine-tune 25 epochs, with initial learning rate set as 0.1 and then decaying by 10 times at epoch 15 and 20. As mentioned in Section 2.2, we report ACL (DS) results using the adversarial branch BN.

As demonstrated in Table 1, while both *Selfie* and ACL (DS) lead to higher precision and robustness compared to the model trained from random initialization, the proposed ACL (DS) yields a significant improvement on [TA, RA] by [**2.14**%, **2.99**%] on CIFAR-10, and [**2.14**%, **3.58**%] on CIFAR-100, respectively. Those gains are significant, especially considering that the pre-training approach in [1] already outperformed all previous state-of-the-arts robust training approaches. By surpassing [1], ACL (DS) establishes the new benchmark TA/RA numbers on CIFAR-10 and CIFAR-100.

We then apply ACL (DS) on Wide-Resnet-34-10 [34], which is also a standard model employed for testing adversarial robustness , to verify if the proposed pre-training can also help for big models. In the supervised fine-tuning experiment with CIFAR10, the random initialization (which equals TRADE[29]) yields [84.33%, 55.46%][1]. By simply using the ACL (DS) pretrained model as the initialization, we obtain [TA, RA] of [85.12%, 56.72%], where our pre-training also contributes to a nontrivial [0.79%, 1.26%] margin for TRADE on the big model.

We further demonstrate that our robustness gain can consistently hold against unforeseen attacks, by leveraging the 19 unforeseen attacks used by [32] on CIFAR-10 with ResNet-18. As shown in

Table 2: Comparison of semi-supervised performance on CIFAR-10 where only 10% (first row) and 1% labels (bottom row) are available for different methods: UAT++ [11], *Selfie* and ACL (DS).

| Method | vanilla UAT++ | | *Selfie* | | ACL (DS) | |
|---|---|---|---|---|---|---|
| Metric | TA(%) | RA(%) | TA(%) | RA(%) | TA(%) | RA(%) |
| 10% labels | 70.79 | 50.43 | 72.16 | 50.37 | 76.66 | 51.09 |
| 1% labels | 41.88 | 30.46 | 53.54 | 37.75 | 75.66 | 50.67 |

Fig. 2, our approach achieves improvement on most of the unforeseen attacks (except gaussian noise, where the performance drop marginally by 0.23%). Remarkably, the proposed approach leads to an improvement of 2.42% when averaged over all unforeseen attacks.

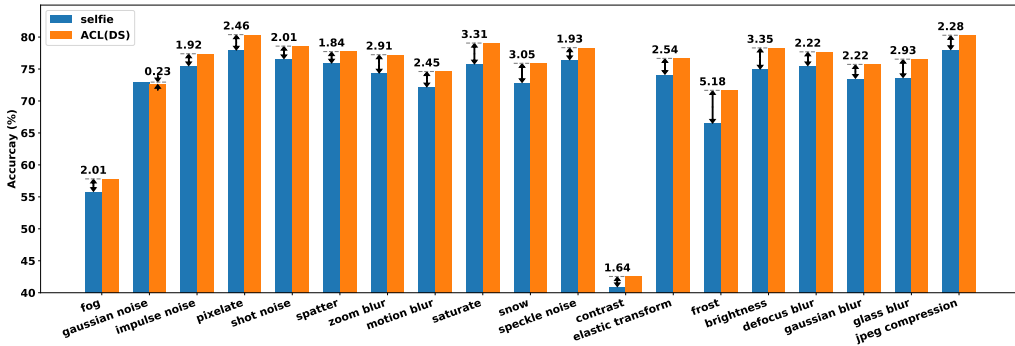

Figure 2: This chart conclude the performance comparison between models fine-tuned from the pre-trained weights with *Selfie* [1] and with ACL (DS) on CIFAR-10. Our proposed ACL (DS) outperforms on most unforeseen attack types, showing more general robustness gains.

### 3.1.2 ACL for semi-supervised adversarial training

Following the procedure in Section 2.2.1, we compare our ACL (DS)-based semi-supervised training, with (1) *Selfie* [1], following the identical routine as our method; and (2) the state-of-the-art UAT++, described in [11]. We emphasize that both *Selfie* and our method are plugged in to improve over the semi-supervised framework like UAT++ (described Sec. 2.2.1), by supplying pre-training, rather than re-inventing a new different semi-supervised algorithm. The results are summarized in Table 2.

We first follow the setting in [11] to use 10% training labels together with the entire (unlabeled) training set. We observe that, adopting *Selfie* as pre-training can improve TA by 1.37% with RA almost unchanged, compared to the vanilla UAT++ without any pre-training. That endorses the notable value of pre-training of data-efficient adversarial learning. Moreover, along the same routine, ACL (DS) outperforms *Selfie* by large extra margins of [4.50%, 0.72%] in [TA ,RA], manifesting the superior quality of our proposed pre-training over [1]. Our result is also the new state-of-the-art TA-RA trade-off achieved in this same setting, as we are aware of.

Encouraged by the promising data efficiency indicated above, we explore a challenging extreme setting that no previous method seems to have tried before: only using **1%** labels together with the unlabeled training set. All methods' hyperparameters are tuned to our best efforts using grid search. As shown in Table 2 bottom row, the advantage of using ACL (DS) pre-training becomes even more remarkable, with [TA, RA] only minorly decreased [1.00%, 0.42%] in this extreme case, compared to using 10% labels. In comparison, both vanilla UAT++ and *Selfie* suffer from catastrophic performance drops (~13% - 30%). This result points to the tremendous potential of boosting the label efficiency of adversarial training much higher than the current.

To further understand why our proposed method leads to such impressive margins, we check the accuracy of pseudo labels generated in *step ii*. We find that ACL (DS) leads to much higher standard accuracy of 86.73% (computed over the entire training set) even with **1%** labels, while vanilla UAT++ and *Selfie* can only achieve 37.67% and 46.75%, respectively. This echos the previous finding in [13] that the quality of pseudo labels constitutes the main bottleneck for semi-supervised adversarial training; and it indicates that our proposed robust pre-training inherits the strong label-efficient learning capability from contrastive learning [2].

Table 3: Performance comparison of pre-trained model in terms their achieved TA-RA under supervised adversarial fine-tuning: Random Initialization (RI), SimCLR(S2S), ACL(A2A), ACL(A2S), ACL (DS) under adversarial training.

| Metric | RI | SimCLR (S2S) | ACL (A2A) | ACL (A2S) | ACL (DS) |
|--------|-------|--------------|-----------|-----------|----------|
| TA(%)  | 79.90 | 81.57        | 78.68     | 80.82     | 82.19    |
| RA(%)  | 49.36 | 49.97        | 50.12     | 52.11     | 52.82    |

Table 4: The adversarial fine-tuning performance comparison of pre-trained model with different BN options for ACL(A2S) and ACL (DS)

| Metric | ACL (A2S) | | | ACL (DS) | | |
|--------|----------|-------------------|-----------------------|----------|-------------------|-----------------------|
|        | SingleBN | DualBN($\theta_{bn}$) | DualBN($\theta_{bn^{adv}}$) | SingleBN | DualBN($\theta_{bn}$) | DualBN($\theta_{bn^{adv}}$) |
| TA(%)  | 75.15    | 81.54             | 80.82                 | 73.15    | 82.46             | 82.19                 |
| RA(%)  | 35.60    | 51.59             | 52.11                 | 40.10    | 52.36             | 52.82                 |

## 3.2 Ablation study for ACL

**Comparing A2A, A2S and DS** We now report the ablation study among our proposed three ACL options, with two extra natural baselines: SimCLR [2] that embeds no adversarial learning (a.k.a, Standard-to-Standard, or S2S); and random initialization (RI) without any pre-training.

As shown in Tab. 3, while SimCLR (S2S) improves TA over random initialization (RI) by 1.67%, it has little impact on RA. That is as expected, since this pre-training considers no adversarial perturbations. A2A boosts RA marginally more than S2S thanks to introducing adversarial perturbations, but pays a dramatic price of TA (even 1.22% lower than no pre-training). Similar to our previous conjecture, it shows that A2A's overly aggressive, worst-case only consistency enforcement degrades the feature quality. A2S starts to achieve more favorable RA, but its TA is still slightly below S2S. Eventually, DS jointly optimize both standard and adversarial feature consistencies, ensuring the feature quality and robustness simultaneously. Not only DS surpassed SimCLR by 2.85% RA, but more interestingly, it also outperforms SimCLR by 0.62% TA. That suggests DS to be a potentially more favored pre-training, even for standard training as well. We consider this to align with another recent observation that adversarial examples as data augmentation can improve standard image recognition too [27]: we leave it to be more thoroughly examined for future work.

**Comparing single and dual BNs** For A2S and S2S, we compare three scenarios: 1) using one single BN for both standard and adversarial branches; 2) using dual BN for training, and adopt the standard branch BN (denoted as $\theta_{bn}$) for testing; 3) using dual BN for training, and adopt the adversarial branch BN ($\theta_{bn^{adv}}$) for testing. As shown in Tab. 4, both methods see notable performance drops when using single BNs that mix standard and adversarial feature statistics, which align with the findings in [27, 28]. For dual BN, our default choice of $\theta_{bn^{adv}}$ leads to more favored RA meanwhile still strong TA. Otherwise if we switch to $\theta_{bn}$, TA performance will be emphasized to become higher, while RAs drop slightly yet remain competitive.

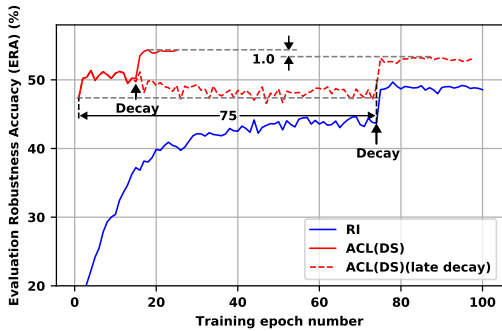

Figure 3: The robust accuracy in cross-validation dataset w.r.t. different epochs. We compare models trained from random initialization (RI) and from ACL (DS). Two different fine-tuning learning rate schedules of ACL (DS) are included.

**Linear separability of pre-trained representations** To evaluate the learned representations, [2] adopted a *linear separability* evaluation protocol, where a linear classifier is trained on top of the frozen pre-trained encoder. The test accuracy is used as a proxy for representation quality: higher accuracy indicates features to be more discriminative and desirable. Here we adapt the protocol by also adversarially training the linear classifier, and

Table 5: Evaluating the performance of the self-supervised representation trained with ACL (DS) by fixing the trained encoder and only fine-tuning the final linear FC layer. Two fine-tuning strategies for two different BNs ($\theta_{bn^{adv}}$ and $\theta_{bn}$) are considered.

| BN choice | $\theta_{bn}$ | | $\theta_{bn^{adv}}$ | |
|---|---|---|---|---|
| Metric | TA(%) | RA(%) | TA(%) | RA(%) |
| Standard Training | 91.02 | 0.20 | 79.76 | 30.31 |
| Adversarial Training | 82.63 | 0.28 | 74.22 | 44.22 |

evaluate the features learned by ACL (DS) pre-training in terms of both TA and RA. We call the RA obtained by the adversarially trained linear classifier as *adversarial linear separability* of the features.

Tab. 5 compares on two different ways training the linear classifier, and also on employing either set of BNs ($\theta_{bn}$ or $\theta_{bn^{adv}}$) to output features. A good level of adversarial linear separability, i.e., 44.22% RA, is observed under $\theta_{bn^{adv}}$. Interestingly, if we switch from $\theta_{bn^{adv}}$ to $\theta_{bn}$, we immediately lose the adversarial linear separability (close to 0% RA). That again confirms the distinct statistics captured by two BNs [28, 27].

**Robustness dynamics during adversarial fine-tuning**   Lastly, we dissect how the robustness grows during the supervised tuning process, when starting from random initialization and ACL (DS) pre-training, respectively. As indicated in Fig. 3, the robust accuracy of ACL (DS) pre-trained models jumps to 47.38% after one epoch fine-tuning, while it costs randomly initialized models 74 epochs more to achieve the same. Additionally, if we fine-tune the ACL (DS) pre-trained model longer and do not anneal the learning rate earlier, we will also see the robust accuracy decrease before the decay point and end up with 1.0% robustness drop, which was reported in [35] and termed as the adversarial over-fitting phenomenon.

# 4   Related work and discussions

ACL demonstrates the value of contrastive-style unsupervised pre-training, for adversarially robust deep learning. Our methodology is deeply rooted in the recent surge of ideas from self-supervised learning (esp. as pre-training [1]), contrastive learning (esp. [2]), and adversarial training (esp. the unlabeled approaches [11]). We review those relevant fields to gradually unfold our rationale.

## 4.1   Pre-training and self-supervision help adversarially robust learning

Training a supervised deep network from scratch requires tons of labeled data, whose annotation costs can be prohibitively high. One promising option is to first pre-train a feature representation using unlabeled data, which is later fine-tuned for down-stream tasks using (few-shot) labeled data. In general, pre-training is discovered to stabilize the training of very deep networks [36, 37], and usually improves the generalization performance, especially when the labeled data volume is not high. Such *pre-training & fine-tuning* scheme further facilitates general task transferability and few-shot learnability of each down-steam task. Therefore, pre-trained models have gained popularity in computer vision [38], natural language processing [39] and so on.

Unsupervised pre-training initially adopted the reconstruction loss or its variants [37, 40]. Such simple, generic loss learns to preserves input data information, but not to enforce more structural priors on the data. Recently, self-supervised learning emerges to train unsupervised dataset in a supervised manner, by framing a supervised learning task in a special form to predict only a subset of input information using the rest. The self-supervised task, also known as *pretext* task, guides us to a supervised loss function, leading to learned representations conveying structural or semantic priors that can be beneficial to downstream tasks. Classical pretext tasks include position predicting [41], order prediction [42, 43], rotation prediction [44], and a variety of other perception tasks [45, 46].

Since adversarially robust deep learning [9] is more data-demanding than standard learning [10], it is natural to leverage unlabeled data using self-supervised learning too. Among other options [14], the latest work [1] adopted the *pre-training & fine-tuning* scheme, which is plug-and-play for different down-stream tasks: we hence choose to follow this setting.

### 4.2 Contrastive Learning brings feature consistency that fits adversarial robustness

Handcrafted pretext tasks [42, 43, 41, 44] largely rely on heuristics, which can limit the generality of the learned representations. An emerging family of self-supervised methods leverages a *contrastive loss* for learning representations, and have shown promising results. The contrastive loss is typically defined for minimizing some distance between positive pairs while maximizing it between negative pairs, which allows the model to learn representations without reconstructing the raw input.

The use of contrastive loss for representation learning dates back to [47], and has been widely studied since then [48, 3, 4, 5, 6, 7, 49, 2]. As shown in [2], the setup of contrastive tasks via non-trivial data augmentation seems to be one major success factor behind contrastive learning. Other aspects, such as CNN architectures, non-linear transformation network in between CNN and contrastive loss, loss functions, batch size and/or memory bank, can all meaningfully impact the performance of learned representations. By combining these factors, a recently proposed self-supervised learning method, SimCLR [2], show that it can learn representations on par with supervised learning while requiring no human annotation. Essentially, SimCLR empowers the unsupervised feature by encouraging *feature invariance* to specified data transformations. Such feature-level invariance is well-known to be desirable for standard generalization of CNNs, yet is often not met by state-of-the-art CNNs [50, 51]. Therefore, enforcing *consistency* w.r.t. data augmentations has been shown effective for unsupervised learning [52, 53, 54].

Although existing contrastive learning literature [48, 3, 4, 55, 5, 6, 56, 49, 7, 2] discussed their boosts on the standard generalization, we realize that the feature consistency (as a result of contrasting) is valuable for robustness too. One cause of adversarial fragility could be attributed to the non-smooth feature space near samples, i.e., small input perturbations can result in large feature variations and even label change. Enforcing consistency during training w.r.t. perturbations, has been thus shown to immediately help adversarial robustness [11, 30, 12]. A closely relevant work [57] showed that a combination of stochasticity and diverse augmentations, plus a feature divergence consistency loss, improved the robustness and uncertainty estimates of fully-supervised image classifiers.

The above observations constitute our conjecture that contrastive learning could be a superior option for adversarial pre-training to other classical self-supervision tasks. Compared to those pretexts adopted by [1], the feature consistency appears to be more directly targeted to the robustness issue.

## 5 Conclusion

In this work, we leverage the contrastive learning to enhance the adversarial robustness via self-supervised pre-training. Our method is motivated by the cause of adversarial fragility, and we discuss several options to inject adversarial perturbations to reduce it. Through experiments in both supervised fine-tuning and semi-supervised learning settings, we demonstrate the proposed adversarial contrastive learning framework can lead to models that are both label-efficient and robust. Potential future work includes investigating the defense of larger models and datasets [58], and the incorporation of more diverse adversarial perturbations.

### Acknowledgments

This work is supported by a US Army Research Office Young Investigator Award W911NF2010240.

### Broader Impact

Defending machine learning models against adversarial attacks is a crucial component towards the goal of making AI systems more secure and trustworthy. Our proposed framework of Adversarial Contrastive Learning can be a powerful tool to improve model robustness in a data-efficient fashion. It advances the latest achievement in robustness-aware pre-training, and the results further raise the state-the-art bars for both supervised and semi-supervised adversarial training. We expect our techniques to contribute to the grand goal of building more secured and trustworthy AI.

## Footnotes

[1]We run the official code of [29]. The RA drops by 1% compared to what [29] reported, since we test with the PGD attack starting from random initializations, which leads to stronger adversarial attacks.

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
