[Reviews · NeurIPS 2020]

Review 1

Summary and Contributions: The paper incorporates adversarial training into the pre-training step, which makes the pre-training techniques even more robustness-aware. This can be seen as an extension of SimCLR (with the incorporation of adversarial training). ==== Post rebuttal update==== I appreciate the authors' response, which clarifies and answers most of my concerns. I believe with a minor revision, the paper can be accepted to NeurIPS. Therefore, I raised my score to 7, an accept.

Strengths: The idea to introduce adversarial robustness into self-supervised learning is novel. One clear advantage of this is to obtain robust models with unlabeled data, which is more readily available.

Weaknesses: More baselines should be included. The paper mainly compares the proposed method to SimCLR, which is the base model that doesn't consider any adversarial during pre-training. Therefore, the results are somewhat trivial (adversarial pre-training should outperform standard pre-training). - Since this is the first work that proposed adversarial-aware pretraining (as the authors claimed), I think it is appropriate to compare its adversarial robustness to some supervised learning methods (such as [1]) or even incorporate these supervised learning methods into the fine-tuning phase (instead of standard adversarial training). - Also, robustness against unforeseen attacks (l1, l2, etc.) should be evaluated. For standard pre-training methods (e.g. SimCLR), every attack is unseen, but those still show improved robustness. I wonder if the adversarial robustness-aware pre-train method can help the model to be even more robust against other unseen attacks? [1] H. Zhang, Y. Yu, J. Jiao, E. P. Xing, L. E. Ghaoui, and M. I. Jordan, “Theoretically principled trade-off between robustness and accuracy,” in Proceedings of the 36th International Conference on Machine Learning, 2019.

Correctness: The related work section doesn't discuss methods that tackle the adversarial robustness problem (including supervised learning or semi-supervised learning). I believe these methods are important in the discussion and can also act as baselines.

Clarity: There are some typos in the papers. These do not hurt the readability, but still make the paper feel less well written. - Line 138: in in -> in - Line 315: intoduing -> introducing - Line 317: discusses -> discuss - Line 219: to aligned - Line 242: from \theta_bn to \theta_bn,adv -> from \theta_bn,adv to \theta_bn - Line 182: except impulse noise -> except Gaussian noise (according to figure 2) - etc. The proofreading could have been better.

Relation to Prior Work: The authors adequately discussed the related works.

Reproducibility: Yes

Additional Feedback:


Review 2

Summary and Contributions: This work shows that as a way of unsupervised pre-training, contrasting features to random and adversarial perturbations for consistency can benefit robustness-aware pre-training even further. This idea, as naturally motivated by the cause of adversarial fragility, yielded state-of-the-art results for adversarial defense. On CIFAR-10/100, the authors presented well-executed experiment on pretraining (for supervised fine-tuning) and semi-supervised learning (up to very low label rates).

Strengths: The paper’s main idea is easy to follow: extending a recently successful contrastive learning framework SimCLR [2] to adversarial training. While SimCLR is already popular for a number of tasks, exploring its usage for adversarial defense appears to be new and original. The authors explained why SimCLR might be particularly suitable for the goal of adversarial robustness: one cause of adversarial fragility is the lack of feature invariance to small input perturbations, and SimCLR learns representations by maximizing feature invariance under differently augmented views. That makes this paper well motivated and grounded. The main technical part of this paper explores options to formulate the contrastive task. As adversarial perturbation is a very strong and hostile form of data augmentations, how to balance its role with SimCLR’s own random augmentations is an important question to address. The authors compares three options: Adversarial-to-Adversarial (A2A), Adversarial-to-Standard (A2S), and Dual Stream (DS) which includes two contrasting pairs (one standard, the other adverserial) and appeared to the best default option. One more contribution is to extend this framework to semi-supervised learning, following the idea of [9]. The experiments are very thorough, and the results are strong. As pre-training for supervised adversarially robust models, this new method outperforms a very recent self-supervised robust pre-training approach [1] with large margins, making it a new state-of-the-art. It proves by injecting adversarial augmentations, contrastive pre-training indeed contributes to learning data-efficient robust models. As a pre-training way (with no extra data), this method has a plug-n-play nature and can work with many more adversarial defense methods. Their robustness gains can even extend to unforeseen attacks (Figure 2), shown by applying 19 unforeseen attacks by [25]. The authors further demonstrate their pre-training can improve over SOTA semi-supervised adversarial training like UAT++ [9]. In a challenging low-label setting of 1% rate, Table 2 shows this method only see a minor decrease (~1%) compared to 100% full label rate, while other baselines drop 13% - 30%. This result (if reliable?) is amazing and could boost a strong community interest in studying unlabeled data for robust training, e.g., does data-only (with few to no labels) suffice for good robustness?

Weaknesses: I’m in general positive about this paper: contributions are clear, and the motivations/logics/experiments all appear to be thoughtful and convincing. A few nitpicks/suggestions: - I’m intrigued by your very good semi-supervsed results at 1% low label rate. Could you possibly include more comparison methods, and could you elaborate more why your method is particularly successful compared to others? - It would be nice to include more backbones besides ResNet18: currently that is the only one used. - Another great thing to add would be ImageNet-scale experiments, although I understand it very hard/impossible for rebuttal window. But please do consider it, as your method shall likely show benefits on it too.

Correctness: Yes

Clarity: The writeup is excellent, and the explanation is rather clear. Figure 1 helped understand the technical content. Besides, I quite like Section 4 that lays out a well-sorted roadmap, connecting dots from previous literature and clearly linking them to this work.

Relation to Prior Work: Yes

Reproducibility: Yes

Additional Feedback: ----------------------------- I have read the author's rebuttal, and decide not to change my score.


Review 3

Summary and Contributions: The paper presents Adversarial Contrastive Learning (ACL) to combine adversarial training and contrastive representation learning. Empirical experiments verify the effectiveness of the presented approach. -------- The rebuttal is doing a good job. Nonetheless, I also like to point out that although the changes are easy to made in a revision, but the changes are a lot. I have updated my rating from 3 to 5 by considering less of the imprecise presentation in the paper.

Strengths: The presented approach is reasonable and may benefit the field of adversarial robustness. The sets of experiments also seem to be comprehensive.

Weaknesses: First, the paper isn't easy to follow. It requires the readers to have sufficient background in adversarial robustness. Second, the author hides lots of details for the presented approach. Third, the presentations and discussions in the experimental section should be further improved. Details will be provided later.

Correctness: I do not find incorrectness in the paper.

Clarity: First, the author assumes the readers are experts in adversarial robustness, and hence it has lots of presentation flaws. For instance, the author mentions PGD way early in Figure 1. However, the author explains what PGD is until line 96. Another example is Section 2.2.1. In lines 131-132, the author skips all the details of supervised adversarial training and only refers to [1]. Second, the structure of the presentation is messy. Equations (3) explains the semi-supervised loss step iii. However, the author does not provide supervised loss and semi-supervised loss step i and ii. I can hardly follow the presentation. The author also does not discuss what the distilling model is in equation (3). Third, in the experimental section, the author shall consider explaining the evaluation metrics beforehand. For instance, it surprises me when I first see "TA, RA" in lines 176. And later, I realize the definition of "TA, RA" is hidden in Table 1. Another example is the lambda in line 171. Where can I find the definition of lambda? Furthermore, the author shall mention the ablation study for ACL (section 3.2) is performed on supervised contrastive learning.

Relation to Prior Work: The paper provides reasonable discussions on prior work.

Reproducibility: No

Additional Feedback: In lines 71-72, are the authors revealing their identity?

[Author Response · NeurIPS 2020]

We truly appreciate all reviewers (R1, R2, R3)' valuable comments, which will greatly help improve our final paper. We have addressed all raised questions below. The final version will be carefully polished up.

**1. Concerns about baselines (R1):** For clarifying, the general idea of adversarially robust pre-training was first explored in [1]. This work's main goal is to improve this idea by leveraging contrastive learning, for the first time. We believe the contrastive learning to fit the goal of robustness better than the ad-hoc "pre-text" tasks employed in [1], for which our intuitions were elaborated in Sections 1 and 4 (directly enforcing feature smoothness). Hence naturally, our main comparison subject in the supervised fine-tuning scheme is also [1], which is the most recent SOTA. We also included SimCLR because it is a straightforward baseline , but it is not any main competitor on robustness results

Also, we already followed [1] to employ TRADE [23] in our supervised fine-tuning. We will make this clear in paper.

**2. Unforeseen attacks robustness (R1):** As you can see from Figure. 2, we already tested the robustness against the set of 19 unforeseen attacks (the most standard benchmark for unforeseen robustness) for models pre-trained with ACL (DS). Comparing to [1], our method achieved performance gains on 18 out of 19 attacks. Per your suggestion, we further test the robustness of ACL (DS)-pretrained models, to $\ell_1$ ($\epsilon = \frac{2000}{255}$) and $\ell_2(\epsilon = 0.5)$ attacks, respectively. We find ACL (DS) still leads to higher robustness of [46.65%, 61.70%], compared to [43.14%, 58.56%] when using [1].

**1. Why the performance of 1% low label rate is so good? (R2):** An important reason for our outstanding semi-supervised results is that our pre-training inherits the strong label-efficient learning property from contrastive learning. Thus, in the *stage ii* of semi-supervised training, the generated pseudo label demonstrates higher accuracy of 86.73% even with only 1% labels available. This greatly contributes to the final robustness. In contrast, other pre-training methods like Selfi in Table 2 can only generate pseudo labels with 46.75% accuracy.

**2. More backbones besides ResNet18 (R2):** We further include results on Wide-Resnet-32-10, which is also a standard model employed for testing adversarial robustness. Using this model, on the supervised fine-tuning experiment with CIFAR10, ACL(DS) leads to [TA, RA] of [85.12%, 56.72%], while random initialization yields [84.33%, 55.46%][1]. Our pre-training still contributes to a nontrivial [0.79%, 1.26%] margin. More results will be added to final version.

**3. ImageNet experiments (R2):** We agree ImageNet experiments would be a nice addition to our paper. As you also kindly acknowledged, due to the short rebuttal time , we will try our best to obtain those results for the final version.

**General response to R3:** We thank R3 for suggesting a few useful points in improving our writeup quality. However, **we respectively disagree** that a clear rejection recommendation could be solely grounded on them.

We find that most presentation issues you kindly pointed out can be easily fixed through revision - please see our point-to-point reply below. We are also humbled to note that the other two expert reviewers think quite positively of our writing quality (**"excellent", "rather clear", "well sorted"**, etc.)

- **The "PGD" in Figure 1:** we will replace PGD with another more general word, e.g., "Attack".
- **Skipped details on supervised adversarial training and only refers to [1]**: we will happily add 2-3 lines to explain the loss function, which is essentially training with the same TRADE loss [23].
- **Loss of semi-supervised steps i and ii:** *Step i* is identical to the ACL pre-training in the supervised fine-tuning case (as indicated in lines 133). *Step ii* is the standard training with the vanilla cross-entropy loss. We can make a table to summarize the three steps' loss functions, if that is considered to improve clarity.
- **The distilling model of semi-supervised step iii :** We employ the standard distilling model with temperature, which was introduced in (Hinton et. al., arxiv May 2015). We will make this clear in paper.
- **Definition of "TA, RA" hidden in Table 1:** No, this is a wrong accusation. Please read lines 149-150 where they appear for the first time and are clearly defined, before they are later used (line 176, table 1, etc.)
- **lambda definition:** Lambda was inherited from [23], and is defined by us in line 146.
- **"Supervised contrastive learning":** Just want to clarify here: "supervised contrast learning" is another different method's name (Khosla et. al., arxiv April 2020), and has nothing to do with our paper. Section 3.2 is on our standard procedure: self-supervised contrastive pre-training, followed by supervised fine-tuning.
- **Revealing identity in line 71-72?** No. What we meant was simply "among the few, now we discuss [9] in more details as it is most relevant to our semi-supervised setting". The word "introduce" might be misleading, and we will revise to "'discuss". We welcome R3 to check back on the identity issue if this paper is accepted.
- **Hiding details for the presented approach?** We are really confused by this critique, and we do not find more details provided by your review on this regard (besides the minor issues that have been addressed above). We are confident that this paper is fully reproducible based on its own content and reference pointers, which has been agreed by the other two reviewers. We also promise to release codes upon acceptance.

## Footnotes

[1]We run the official code of [23] for baseline. The RA drops by 1% compared to [23] reported, since we test with the PGD attack started from random initializations, which leads to stronger adversarial attack. Details will be fully specified in final paper.


[Meta-Review · NeurIPS 2020]

This paper focuses on adversarial training. The proposal is to incorporate adversarial training into the pre-training step, which makes the pre-training techniques even more robustness-aware. This can be seen as an extension of SimCLR (with the incorporation of adversarial training). The philosophy behind sounds quite interesting to me, namely, introducing adversarial robustness into self-supervised learning and formulating the contrastive task. This philosophy leads to a novel algorithm design I have never seen, i.e., Adversarial-to-Adversarial (A2A), Adversarial-to-Standard (A2S), and Dual Stream (DS). The clarity and novelty are clearly above the bar of NeurIPS. While the reviewers had some concerns on the significance, the authors did a particularly good job in their rebuttal. Thus, most of us have agreed to accept this paper for publication! Please carefully address R3' comments in the final version, namely, revising less of the imprecise presentation in the paper.